# Evaluating the Utility of *Simplicillium lanosoniveum*, a Hyperparasitic Fungus of *Puccinia graminis* f. sp. *tritici*, as a Biological Control Agent against Wheat Stem Rust

**DOI:** 10.3390/pathogens12010022

**Published:** 2022-12-23

**Authors:** Binbin Si, Hui Wang, Jiaming Bai, Yuzhen Zhang, Yuanyin Cao

**Affiliations:** 1College of Plant Protection, Shenyang Agricultural University, Shenyang 110866, China; 2College of Biological Science and Engineering, North Minzu University, Yinchuan 750021, China

**Keywords:** wheat stem rust, *Puccinia graminis* f. sp. *tritici*, hyperparasitism, *Simplicillium lanosoniveum*, identification

## Abstract

Wheat stem rust is one of the wheat diseases caused by *Puccinia graminis* Pers. f. sp. *tritici (Pgt)*. This disease has been responsible for major losses to wheat production worldwide. Currently used methods for controlling this disease include fungicides, the breeding of stem rust-resistant cultivars, and preventive agricultural measures. However, the excessive use of fungicides can have various deleterious effects on the environment. A hyperparasitic fungus with white mycelia and oval conidia, *Simplicillium lanosoniveum*, was isolated from the urediniospores of *Pgt*. When *Pgt*-infected wheat leaves were inoculation with isolates of *S. lanosoniveum*, it was found that *S. lanosoniveum* inoculation inhibited the production and germination of urediniospores, suggesting that *S. lanosoniveum* could inhibit the growth and spread of *Pgt*. Scanning electron microscopy revealed that *S. lanosoniveum* could inactivate the urediniospores by inducing structural damage. Overall, findings indicate that *S. lanosoniveum* might provide an effective biological agent for the control of *Pgt*.

## 1. Introduction

Eight major cereal crops, wheat, rice, barley, rye, oats, corn, sorghum, and millet, provide two-thirds of the world’s food supply. Each of these major cereal crops is vulnerable to at least one rust disease. Rust pathogens can induce substantial damage to cereal crops, reduce yields, and cause major economic losses [1]. Wheat stem rust poses a major threat to wheat production worldwide because outbreaks of this disease can occur over large spatial scales and spread rapidly [2]. Wheat stem rust infections affect the stems and leaves of wheat plants. In severe cases, wheat stem rust can reduce wheat yield by 75% or result in the death of host plants [3].

Hyperparasitic fungi parasitize other plant pathogens, and both hyperparasitic fungi and their hosts impose mutual constraints on each other’s growth [4]. Hyperparasitic fungi are widespread in natural ecosystems and commonly parasitize filamentous fungi [5]. These fungi might provide promising alternatives to chemical fungicides for the control of fungal diseases in plants [6]. They have thus received much research attention from botanists for their potential to be used as biocontrol agents against plant pathogens. To date, hyperparasitic fungi have been used to control a few types of rust diseases and powdery mildew [7]. Following the colonization of host pathogens, the hyphae of hyperparasitic fungi grow toward the edge of the colony. Once the hyphae infiltrate the cell wall of the host to obtain nutrients, a parasitic relationship is established between the hyperparasitic fungus and the host pathogen, which can induce damage to the pathogen. Many hyperparasitic fungi occur naturally in plants in the form of mold colonies.

Hyperparasitic fungi effectively prevent plant pathogens from infecting their hosts; these interactions between the plant pathogen and hyperparasite can inhibit the reproductive ability of the pathogenic organism [8,9]. The antagonism between plant and parasitic fungi and that between parasitic fungi and their hyperparasitic fungi can inhibit the growth and reproductive capacity of pathogenic organisms and thus reduce the severity of plant diseases [8]. Rapidly growing hyperparasitic fungi might have a lower impact on the environment compared with chemical fungicides, given that they are ubiquitous in natural ecosystems. This observation has inspired much interest in the development of strategies to use these fungi as biological control agents [9]. 

Many biological antagonists have been shown to work effectively against rust fungi. The application of hyperparasitic fungi is potentially an effective strategy to prevent plant diseases. A total of 30 hyperparasitic fungal genera parasitize rust fungi, including Fusarium (*Fusarium* spp.), *Cladosporium*, *Scytalidium uredinicola*, and *Tuberculina* [10,11]. Parasitism of the urediniospores of rust fungi has been reported in various fungal species [12], including *C. cladosporioides*, *Eudarluca caricis*, *Microdochium caricis*, *Microdochium nivale*, *Lecanicillium lecanii*, and *Alternaria alternata*. Several *Cladosporium* species, such as *C. uredinicola*, *C. cladosporioides*, *C. pseudocladosporioides*, and *C. sphaerospermum*, have been reported to parasitize fungi in the order *Pucciniales* [13].

Scanning electron microscopy (SEM) of the abnormal urediniospores of wheat stripe rust revealed that they were parasitized by *Cladosporium* spp., which inactivated the urediniospores and induced structural damage [14]. Multiple hyperparasitic fungi have been isolated from infected branches of *Pinus armandii*, and the defense mechanisms of fungi against *Cronartium ribicola* have been characterized using SEM [15]. The hyperparasitic fungus *Alternaria alternata* was first reported to parasitize wheat stripe rust in 2017, and its potential utility as a biocontrol agent was demonstrated in a previous study [5].

Current methods used for the control of stem rust include the application of fungicides, the use of stem rust-resistant varieties, and preventive agricultural measures [16]. However, the application of nonselective fungicides can lead to a disease more difficult to manage [9]. In addition, the heavy use of fungicides can have deleterious effects on the environment, along with animals and humans [6]. Here, the utility of hyperparasitic fungi for the control of *Pgt* infection is evaluated. Our findings indicate that hyperparasitic fungi could provide an effective tool for controlling destructive rust diseases of major cereal crops. The isolates of the hyperparasitic fungi in our experiment effectively inhibit the growth of *Pgt* and will aid future studies aimed at the development of environmentally friendly biological agents.

## 2. Materials and Methods

### 2.1. Isolation and Purification of Hyperparasitic Fungi

We were multiplying *Pgt* urediniospores and conducting various experiments with urediniospores in our laboratories at Shenyang Agricultural University, Shenyang. Hyperparasitic fungi were isolated from the urediniospores of *Pgt*, which were cultured at 22 °C and 60% relative humidity in an artificial climate chamber. The susceptible wheat cultivar Little Club was inoculated with *Pgt* (34C3RTGQM) and then placed in an artificial climate chamber, with 60% relative humidity, temperatures kept at 22 °C, and a 16 h/8 h light/dark photoperiod.

The white hyphae were collected from the urediniospores of *Pgt*-infected wheat leaves, inoculated onto potato dextrose agar (PDA) plates using an inoculation loop following 16 d of culture, and then incubated in an inverted position at 25 °C for 7–10 d in a biochemical incubator. After extracting fresh hyphae from the edge of the fungal colony, they were transferred to fresh sterile PDA medium and incubated under the same conditions to obtain a pure culture. Five replications of these experiments were performed. The purified isolates of hyperparasitic fungus were preserved on the PDA plate.

### 2.2. Morphological Observations of Hyperparasitic Fungi

Hyphae with vigorous growth at the edge of the isolated and purified colonies were transferred to fresh sterile PDA medium and placed in an incubator with a 12 h/12 h dark/light photoperiod at 25 °C for 7 d to make observations of the morphology of its colonies, hyphae, and conidia. Observations of hyphal and conidial morphology were performed using an Olympus BX51 microscope.

### 2.3. SEM of Stem Rust Fungi and Hyperparasitic Fungi 

Leaves inoculated with only *Pgt* urediniospores were used as the CK. The hyperparasitic of isolate fungus was placed into a fungal spores suspension (10^5^–10^6^/mL), which was sprayed on the *Pgt*-infected wheat leaves. One day after *Pgt* inoculation, the *Pgt*-infected leaves were inoculated with a suspension of isolate fungus, and kept in the artificial climate chamber for growth under the same conditions. To clarify *Pgt*–parasite interactions during infection, the *Pgt*-infected wheat leaves of hyperparasitic were cut into blocks (0.2 cm × 0.2 cm) and stored in fixative for SEM. The leaves were sampled at 0, 1, 3, 5, 7, and 9 d after smearing the hyperparasitic fungal suspension.

### 2.4. Molecular Identification of Hyperparasitic Fungi

Purified hyphae (0.1 g) of hyperparasitic fungus were placed into 2.0 mL centrifuge tubes. The OMEGA (USA) HP Fungal DNA Kit was used to extract DNA from samples. A NanoDrop Microvolume Spectrophotometer (Thermo) was used to determine the concentration of DNA. Polymerase chain reaction (PCR) was then performed to amplify the internal transcribed spacer 1 (ITS1) and internal transcribed spacer 4 (ITS4) sequences (ITS1 primer: 5′-TCCGTAGGTGAACCTGCG-3′; ITS4 primer: 5′-TCCTCCGCTTATTGATATGC-3′) [17,18]. Electrophoresis with a 1.0% agarose gel was performed to analyze the PCR-amplified products. The thermal cycling conditions were as follows: denaturing at 94 °C for 5 min; 30 cycles (from 94 °C for 35 s, 52 °C for 60 s, to extension at 72 °C for 90 s); to extension at 72 °C for 10 min; and 4 °C for 5 min. PCR products were separated by 1% gel electrophoresis. The extracted DNA samples of the hyperparasitic fungi of *Pgt* were sent to Sangon Biotech Co. (Shanghai, China) for sequencing and molecular identification.

### 2.5. Phylogenetic Analysis

The sequences of Simpliciium spp. were downloaded from the NCBI and aligned by ClustalW in software MEGA10 using the default parameters [19]. The fungal isolates were clustered based on their ITS sequences using the neighbor-joining (NJ) method using MEGA10, and the branch robustness was determined using 1000 bootstrap replications [19].

### 2.6. Hyperparasitic Fungal Inhibition Experiment

The susceptible wheat (Little Club) was used for propagating *Pgt* urediniospores. When the first leaf had expanded after 7 days, seedlings were inoculated with 34C3RTGQM, a predominant race of *Pgt* in China. The collected urediniospores of *Pgt* race 34C3RTGQM were diluted with water to 25 mg·mL^−1^ and inoculated by brush. The *Pgt*-inoculated plants were incubated in an artificial climate chamber at 22 °C with 16 h light photoperiod. The hyperparasitic of fungus was placed into a fungal spores suspension (10^5^–10^6^/mL), which was sprayed on the *Pgt*-infected wheat leaves. Healthy wheat leaves receiving *S. lanosoniveum* inoculation represented control check1 (CK1). Leaves inoculated with only *Pgt* urediniospores were used as CK2. Three, six, and nine days after *Pgt* inoculation, the plants in different pots were inoculated with suspension (10^5^~10^6^ spores/mL) of hyperparasite, and kept in the artificial climate chamber for growth under the same conditions. Each treatment was carried out with wheat seedlings growing in three independent pots. All treatments were placed in the same artificial climate chamber, and observation of the symptoms was performed at the same time.

### 2.7. Urediniospore Germination Inhibition Experiment

Seedlings of wheat (Little Club) were grown in the artificial climate chamber, waiting until the first leaf had expanded after 7 days, inoculated with urediniospores of *Pgt* (34C3RTGQM), and incubated in an artificial climate chamber at 22 °C. Fungal suspensions (10^5^–10^6^ CFU/mL) were prepared in sterile water with fresh urediniospores and hyperparasitic fungal hyphae, which were extracted with a sterilized inoculating loop. The treatment group comprised a mixture of equal volumes of urediniospore suspension and hyphal suspension. In the control group, the urediniospore suspension was mixed with equal volumes of sterile water and cultured in a biochemical incubator in the dark at 22 °C for 6 h in an artificial climate chamber. Suspensions from the treatment and control groups were transferred to glass slides to perform counts of the germinated urediniospores using a light microscope. Germinating urediniospores of *Pgt* produce a germ tube that is at least half the length of the urediniospores. The germination rate was expressed as a percentage based on 100 randomly selected urediniospores. Experiments were performed in triplicate.

## 3. Results and Analysis

### 3.1. Morphological Observations of Hyperparasitic Fungi

The hyperparasitic fungus was white and slow-growing on the PDA medium (Figure 1A). Following 10 d of incubation at 25 °C, fungal colonies (50–60 mm in diameter) were dense and appeared brownish-orange on the bottom of the dish (Figure 1B). Microscopic observation revealed intertwined hyphae variable in diameter, oval conidia, and spores that were stacked or arranged in rows. The hyphae and conidia of the isolated filamentous fungus were highly similar to those of Simplicillium spp.

### 3.2. Molecular Identification of Hyperparasitic Fungi

The hyperparasitic strain IS698-2 was isolated from *Pgt* urediniospores. This hyperparasitic strain of IS698-2 was sequenced and identified as Simplicillium lanosoniveum.

Phylogenetic analysis was performed on Simplicillium. Alignments of the gene sequences were performed using data and tools available in the National Center for Biotechnology Information database. The phylogeny of fungal ITS sequences was built using the neighbor-joining (NJ) method with 1000 bootstrap replicates in MEGA10software (Figure 2).

### 3.3. Pathogenicity Test to Confirm Hyperparasitism: S. lanosoniveum Parasitizing Pgt 

The hyperparasitic ability of *S. lanosoniveum* was confirmed by pathogenicity testing (Figure 3). Fungal suspensions of *S. lanosoniveum* with spore concentrations of 10^5^–10^6^/mL were evenly sprayed on wheat leaves without any symptoms (Figure 3A). Leaves inoculated with only *Pgt* urediniospores produced abundant urediniospores after 12 days post-inoculation (Figure 3B, C). The suspensions of *S. lanosoniveum* inoculated with *Pgt*-infected wheat leaves were observed at 3, 6, and 9 dpi (Figure 3D, E, F); 3 and 6 dpi were without any symptoms (Figure 3D, E), but at 9 dpi, white hyphae wrapped the pustule (Figure 3F).

The production of urediniospores ultimately ceased, suggesting that S. lanosoniveum inhibited the urediniospore production of *Pgt*. This also suggests a possible parasitic relationship between S. lanosoniveum and *Pgt*.

### 3.4. Urediniospore Production Inhibition Experiment

The experiment was performed in triplicate, and average values were used in the analysis. The germination rate of urediniospores in the treatment group was 17% (17 germinated urediniospores out of 100 total urediniospores examined); the germination rate in the control group was 91% (i.e., 91 germinated urediniospores out of 100 total urediniospores) (Figure 4 and Figure 5). These findings demonstrated that *S. lanosoniveum* can inhibit the germination of *Pgt* urediniospores.

### 3.5. SEM of Stem Rust Fungi and Hyperparasitic Fungi

SEM was used to study the interaction between *S. lanosoniveum* and the urediniospores of *Pgt*. Significant morphological changes in the spores and hyphae of *S. lanosoniveum*-parasitized *Pgt* were observed; specifically: conidia were oval, and multiple spores were stacked or in groups (Figure 6D,E). Under normal conditions, the urediniospores of *Pgt* were obovoid or oval (Figure 6A,B). During the initial stage of infection by hyperparasitic fungi, *Pgt* urediniospores showed few signs of parasitization (Figure 6C,D). The attachment of *S. lanosoniveum* spores to urediniospores was observed during the middle stage of infection (Figure 6E). During the final stage of infection, the urediniospores were completely colonized by hyphae and spores of *S. lanosoniveum* (Figure 6F).

Changes in the morphology of *Pgt* urediniospores suggested that *S. lanosoniveum* parasitizes *Pgt* during the sporulation stage, which inactivates the urediniospores. The spores of *S. lanosoniveum* began to germinate after contact was made with the urediniospore surface, and the urediniospores of *Pgt*. were penetrated with its hyphal germ tube. Following the invasion and colonization of urediniospores, hyperparasites likely absorbed nutrients from urediniospores, which decreased the viability of urediniospores and caused them to shrink. The hyphal and conidial morphology of this hyperparasitic fungus was similar to that of members of the genus *Simplicillium*.

## 4. Discussion

Studies on the parasitism of cereal pathogenic fungi by other fungi are important for the development of biological agents for the control of diseases in economically important cereal crops. The identification of hyperparasitic fungi is critically important for understanding their biodiversity. Several hyperparasitic fungi that can parasitize plant rust pathogens have been reported in previous studies, such as *Aphanocladium album, Fusarium* spp., *Lecanicillium* spp., and *Scytalidium uredinicola* [9]. Fungal parasitism has been suggested to be an effective strategy for the control of several diseases. For example, *Ampelomyces quisqualis* has been used to control powdery mildew on grapes and other crops [20]. *Trichoderma* spp. have been used to mitigate the effects of *Fusarium oxysporum* on tomato plants [21]. However, few studies have examined the efficacy of using hyperparasitic fungi for the control of *Pgt* on cereal crops. In our study, we isolated a fungus that could parasitize the urediniospores of *Pgt*. This hyperparasitic fungus could thus potentially be used for the control of *Pgt*, given that it inhibits the development of the urediniospores of *Pgt*.

We studied the morphology of this hyperparasitic fungus, obtained ITS sequences, and built a phylogeny using the NJ method in MEGA10. The molecular data indicated that this fungus belonged to the genus *Simplicillium*, and the identification of this fungus as *S. lanosoniveum* had 99% bootstrap support [22,23].

The most practical approaches for controlling wheat stem rust that are currently used include the breeding of stem rust-resistant varieties and the application of fungicides. The loss of fungal-specific resistance in wheat varieties and the potential resistance of pathogens to fungicides are some of the major challenges in the development of biological agents for the control of wheat stem rust.

## 5. Conclusions

The present study identified *S. lanosoniveum* as a new hyperparasite of *Pgt*. Characterizing the effects of hyperparasitic fungi on plant pathogens is essential for evaluating their utility as biological agents for the control of plant pathogenic fungi. *S. lanosoniveum* inhibited *Pgt* infection through its inhibitory effects on the development and survival of the urediniospores of *Pgt*. However, additional studies are needed to determine whether *S. lanosoniveum* can be used as a biocontrol agent for stem rust under field conditions. More studies are also needed to evaluate the environmental impact of hyperparasitic fungi and their potential to be used for the control of other rust pathogens.

## Figures and Tables

**Figure 1 pathogens-12-00022-f001:**
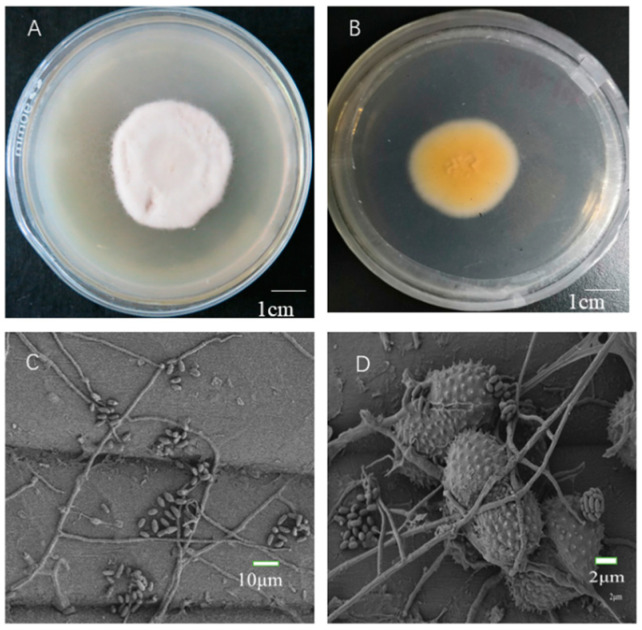
(**A**,**B**) show the morphology of the spores and hyphae of the hyperparasitic fungus on the top and bottom of the dish; (**C**,**D**) show the characteristics of hyphae and conidia in morphology under a scanning electron microscope (×500, ×2000).

**Figure 2 pathogens-12-00022-f002:**
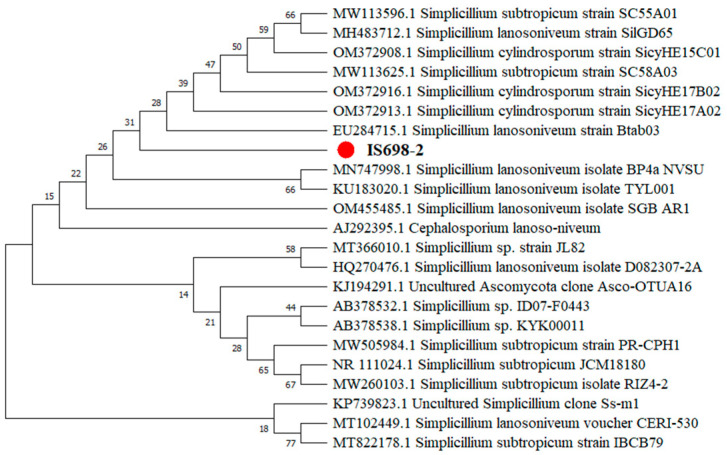
Phylogenetic tree of Simplicillium constructed using the NJ method. The red dot indicates the hyperparasitic *S. lanosoniveum* isolated in this study.

**Figure 3 pathogens-12-00022-f003:**
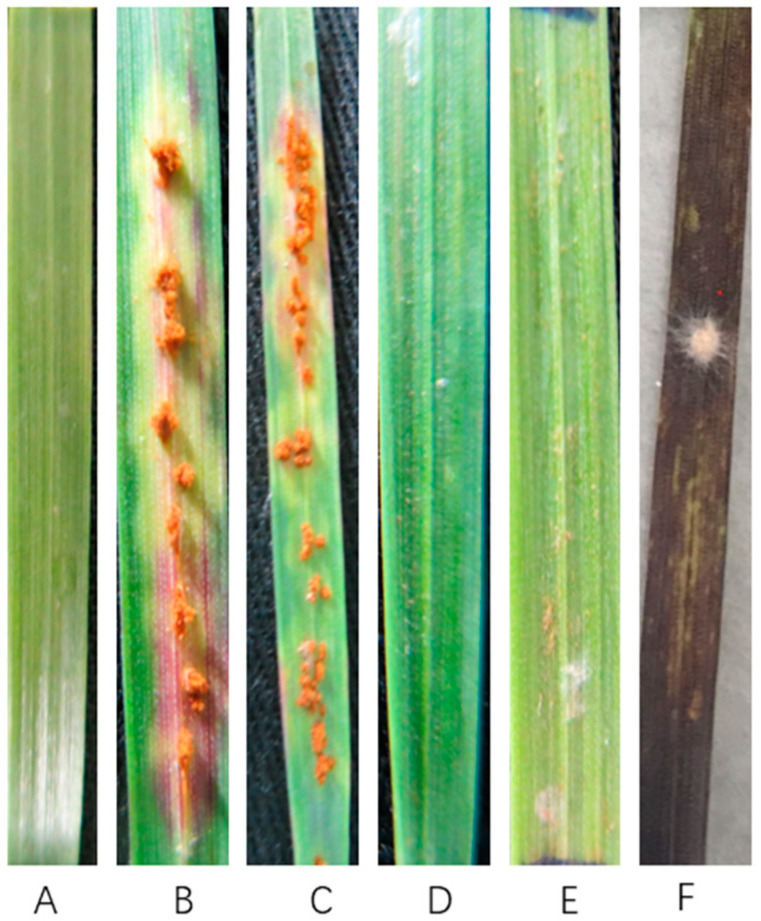
*The pathogenicity test to confirm that S. lanosoniveum could hyperparasitize Pgt*. (**A**): CK1, wheat leaves (cultivar: Little Club) inoculated with *S. lanosoniveum*, 12 dpi, without symptoms; (**B,C**): CK2, symptoms on wheat leaves (Little Club) inoculated only with *Pgt*, 12 dpi; (**D,E**): wheat leaves sprayed with *S. lanosoniveum* suspension following inoculation with *Pgt* at 3 and 6 dpi. No urediniospores were observed on the leaves at 12 dpi; (**F**): pustules on wheat leaves inoculated with *Pgt* at 9 dpi (same cultivar and race as above), and white hyphae wrapped pustules inoculated at 15 dpi with *S. lanosoniveum* suspension.

**Figure 4 pathogens-12-00022-f004:**
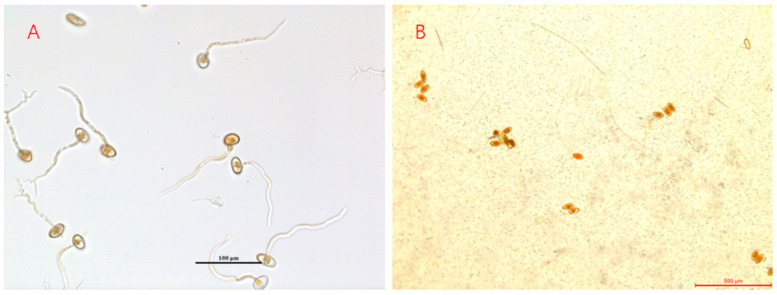
(**A**) shows Pgt urediniospores cultures in water; (**B**) shows the cultures of Pgt urediniospores and *S. lanosoniveum* in water.

**Figure 5 pathogens-12-00022-f005:**
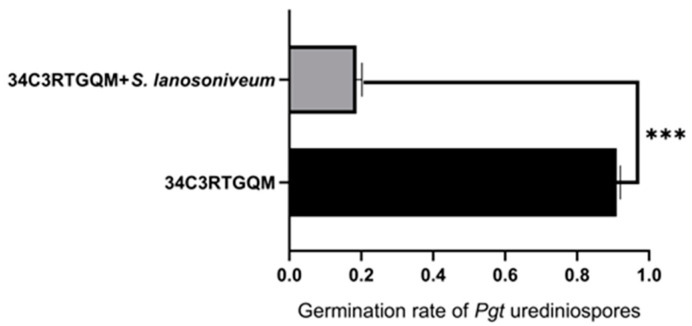
The germination rate of the mixed culture of hyperparasitic fungi (*S. lanosoniveum*) and *Pgt* urediniospores, and the germination rate of *Pgt* urediniospores in water. Data are the means of three independent replicates. *** *p* ≤ 0.001.

**Figure 6 pathogens-12-00022-f006:**
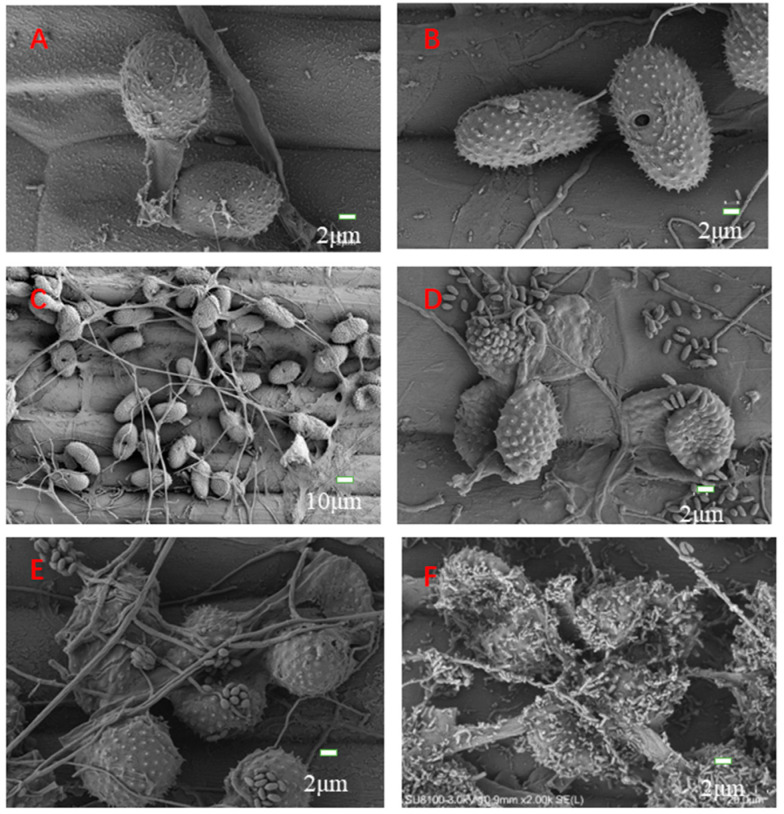
SEM of *Pgt*-infected wheat leaves infected with hyperparasitic fungi (*S. lanosoniveum*). (**A**). CK, wheat leaves (Little Club) inoculated only with *Pgt* urediniospores. (**B**). Hyphal growth of hyperparasitic fungi observed on the surface of urediniospores of *Pgt* at 1 dpi. (**C**). Germ tubes formed in the hyperparasitic fungi at 3 dpi. (**D**). *Pgt* urediniospores at the middle stage (5 dpi) of infection. (**E**). *Pgt* urediniospores partly covered by hyperparasitic fungal spores at 7 dpi. (**F**). The complete colonization of urediniospores of *Pgt* by hyperparasitic fungi at 9 dpi.

## Data Availability

Not applicable.

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
