# Peer review of "Evaluating the Utility of *Simplicillium lanosoniveum*, a Hyperparasitic Fungus of *Puccinia graminis* f. sp. *tritici*, as a Biological Control Agent against Wheat Stem Rust"

_pathogens, 2022, doi:10.3390/pathogens12010022_

Round 1

Reviewer 1 Report

The paper deals with a very important problem – biological control of such devastating disease of wheat as stem rust. The use of hyperparasitic fungi for controlling rusts may be a promising way of plant protection, especially in organic farming.

The authors succeeded in isolating the hyperparasitic fungus from Puccinia graminis urediniospores and identified it as Simplicillium lanosoniveum.

I recommend some revisions 

Lanes 65-67 – please check the appropriateness of reference [14] as in this paper Cladosporium ramosum is not mentioned.

The details of procedures used in the study are given in 2. Materials and Methods and they should not be repeated in 3. Results and analysis, in particular in lanes 155, 156, 185-190.

Please supplement 2. Materials and Methods with the subsection ‘2.7. Data analysis’ and transfer there the part of the text from 3.2 (lanes 159-162) “Alignments of the gene 159 sequences were performed using data and tools on the National Center for Biotechnology 160 Information database. The phylogeny of fungal ITS sequences was built using the 161 neighbor-joining (NJ) method with 1,000 bootstrap replicates in MEGA7 software”

In 3.2 describe the results of phylogenetic analysis (the tree) in more detail to prove that your fungus is indeed Simplicillium lanosoniveum.  Fig.2 does not show what species are close to your strain. Please show names of species on the tree as the numbers only are not sufficient.

In 2.1, please specify the location and time of stem rust sampling, name of the wheat cultivar or line.

In 2.4  lane 119, please add the number of cycles.

In 2.4 and 2.5, please specify the stem rust race used in the experiments.

In 2.5 specify what wheat cultivar was used and how old were the plants from which the leaves were taken for the experiment

2.6. lanes 138, 139 replace ‘urediospores’  by ‘urediniospores’

 In the manuscript there is no figure with the electrophoregram of the amplicon, so remove “(Figure 2)” from lane 157.

In 3.3. there is a discrepancy with 2.5 in the procedure of applying S. lanosoniveum onto leaves: in 2.5. the leaves were “smeared with the suspension......using a ....brush”, but  in 3.3. “ Fungal suspensions of S. lanosoniveum ...were evenly sprayed”

3.3. and 3.4 (about urediniospore germination) have the same titles “Urediniospore production inhibition experiment”

There is also a discrepancy between conditions of incubation of fungal suspensions in 3.4 and 2.6: in 2.6, lane 136, it was incubation in the dark at 25ºC for 6 h, but in 3.4  it was  at 22ºC for 12 h (lane 188).

Fig. 3 lane 180: what is 34C3RTGQM? If this is a name of the race, mention this race in 2. Materials and methods

lane 16   replace ‘Simplicillum’  by ‘Simplicillium’

lane 153 replace ‘electorn’  by ‘electron’

Fig.5 The right portion of the diagram should be removed and there are some problems with the legend on the right (evidently black is Control)

Fig.4. Maybe “A shows.... B shows”

Reviewer 2 Report

The paper is well written with minor changes recommended. There should be reference to virulence changes in the pathogen that cause loss of resistance. It is not linked to fungicide use. It is also unclear at what stage the fungal suspension were applied to the infected leaves. That should be addressed as it can have a major impact on the interaction.
